# DeepFoids: Adaptive Bio-Inspired Fish Simulation with Deep Reinforcement Learning

**Yuko Ishiwaka**[1][*][†]**, Xiao S. Zeng**[2][*]**, Shun Ogawa**[1]**,**
**Donovan Michael Westwater**[2]**, Tadayuki Tone**[1]**, Masaki Nakada**[2][‡]
[1]SoftBank Corp., Japan, [2] NeuralX Inc., USA

## Abstract

Our goal is to synthesize realistic underwater scenes with various fish species in different fish cages, which can be utilized to train computer vision models to automate fish counting task. It is a challenging problem to prepare a sufficiently diverse labeled dataset of images from aquatic environments. We solve this challenge by introducing an adaptive bio-inspired fish simulation. The behavior of caged fish changes based on the species, size and number of fish, and the size and shape of the cage, among other variables. In this paper, we propose a method for achieving schooling behavior for any given combination of variables, using multi-agent deep reinforcement learning (DRL) in various fish cages in arbitrary environments. Furthermore, to visually reproduce the underwater scene in different locations and seasons, we incorporate a physically-based underwater simulation.

## 1 Introduction

The sustainability of the food supply and the protection of marine resources are two of the most important global issues, and are part of the agenda set by the United Nations to be achieved by 2030 [§]. Fish farming is one of the keys to a sustainable seafood supply to support the global population, as seafood consumption is growing rapidly. One of the current major problems in the fish farming industry is managing feeding. Under-feeding slows down the growth of the fish due to malnutrition. Over-feeding is even worse as it can kill the fish and the residual food contaminates the marine environment. Therefore, we attempt to automate fish counting using inexpensive commercially available RGB cameras and computer vision to optimize the amount and timing of feeding.

Deep learning has achieved great success in the field of computer vision. The quality of the dataset determines the accuracy of deep learning models, but it is difficult to obtain data to train a fish counting model. Ishiwaka et al. [1] proposed a fish schooling simulation called Foids, and demonstrated the efficacy of using a CG synthetic dataset for training. Foids adds rules based on fish biology to the Boids [2] algorithm. However, both Boids and Foids require manually setting a number of parameters for a given fish species, which must be adjusted again should any conditions change. To solve this problem, we propose a method to autonomously generate schooling behavior in fish using multi-agent deep reinforcement learning. Fish behavior in a fish farming cage depends not only on natural factors like temperature and light intensity, but also the size and shape of the fish cage, and even the species, number, and size of the fish themselves. By taking into account biological data such as the preferred light intensity, temperature, and inter-fish distance of each fish, our method achieved several distinct patterns of collective behavior depending on population density through autonomous

---

[*]Yuko Ishiwaka and Xiao S. Zeng equally contributed to this work.

[†]`yuko.ishiwaka@g.softbank.co.jp`

[‡]`masaki@neuralx.ai`

[§]`https://sdgs.un.org/2030agenda`

36th Conference on Neural Information Processing Systems (NeurIPS 2022).

learning. Furthermore, we developed a physically-based underwater environment simulation. This simulation is capable of accurately reproducing the conditions of underwater scenes of arbitrary locations and seasons. The bio-inspired fish simulation and physically-based environment simulation allow for the creation of a high-quality synthetic dataset, with which we successfully trained a deep learning model to count fish of various species in fish cages.

In this paper, we explore related work in Sec. 2. We then describe the implementation details of the simulation, data synthesis, and the computer vision models in Sec. 3 and 4. In Sec. 5, we show the experimental results and analysis followed by a conclusion in Sec. 6.

## 2    Related Work

It is well known that mammals use reinforcement learning [3]. Experiments in neuroscience have demonstrated that dopamine neurons play a role in rewards in reinforcement learning [4, 5, 6]. These dopamine neurons are present not only in mammals but in flies and fish as well [7]. It has been shown that dopamine neurons are used in cooperative behavior [8] and learning [9] in fish. Therefore, we adopted reinforcement learning for training cooperative schooling behavior.

In the field of fish swimming simulations, DRL has been used to produce efficient swimming and verify adaptation behavior [10, 11, 12, 13]. These studies focused on simulations of a few fish or soft-bodied animals, but they did not cover large numbers of fish or differences between species. Tu and Terzopoulos [14] proposed a fish simulation with perception, cognition and muscle based locomotion. Lindsey [15] defined 12 types of locomotion in fish, and Satoi et al.[16] incorporated this definition and proposed a controller which automatically selects a type of locomotion based on a desired target behavior. Although those methods worked well at simulating individual fish behaviors, they were not directly useful for our objective of simulating tens of thousands of fish, while forming schooling behavior organically through interactions among the fish.

DRL-based approaches have also been studied for flocking control tasks [17, 18, 19] and crowd simulation [20, 21]. These models do not take biological specifics and their influences on flocking behaviors of simulated objects into consideration, while the adaptation of these species-dependent characteristics constitute a crucial part of our fish control system.

There are relatively few publicly available computer vision datasets of underwater images. Three major such datasets are OzFish [22], Deepfish [23], and a dataset from Ditria et al. [24]. These datasets lack the species and environments seen in fish farms, making them unsuitable for our goals.

The biological properties we use in our simulation are aggregation, cohesion and separation rules[25, 26]; fish's preferred temperature and light intensity [27, 28, 29]; personal space [30, 31]; sense of depth due to hydrostatic pressure [32, 33]; response to obstacles [34, 35]; and decision making interval [36], as well as the social ranking of fish. Fish often form dominance hierarchies, where the dominant fish display aggressive behavior by chasing the subordinate individuals from time to time [37, 38]. The details of the biological background are described in the Appendix. In this paper, we set rewards and penalties for DRL based on these properties.

## 3    Fish Simulation

In this section, we introduce a bio-inspired fish simulation trained with DRL. The bold letters $\mathbf{v}$, $\mathbf{d}$ and $\mathbf{u}$ are used to denote directional 3D vectors, and the uppercase calligraphic letter $\mathcal{I}$ denotes a higher-order tensor. We structure the behavioral model of fish as a multi-agent reinforcement learning problem, where fish agents need to navigate around a constrained 3D space that resembles a fish farm environment while adapting to multiple factors that simultaneously influence their behavior. During the course of interacting with the environment and other fish, each agent acts based on its own observations and is responsible for learning a general policy that maximizes a reward. We model the policies using neural networks and optimize them using Proximal Policy Optimization (PPO) [39] with Generalized Advantage Estimation (GAE) [40].

A schematic illustration of our fish simulation pipeline is provided in Fig.1. The parameters controlling biological and environmental factors in the fish simulation are set based on the literature presented in the Appendix and field data collected from fish farms at Onmaehama and Nishiki, Japan. Fish

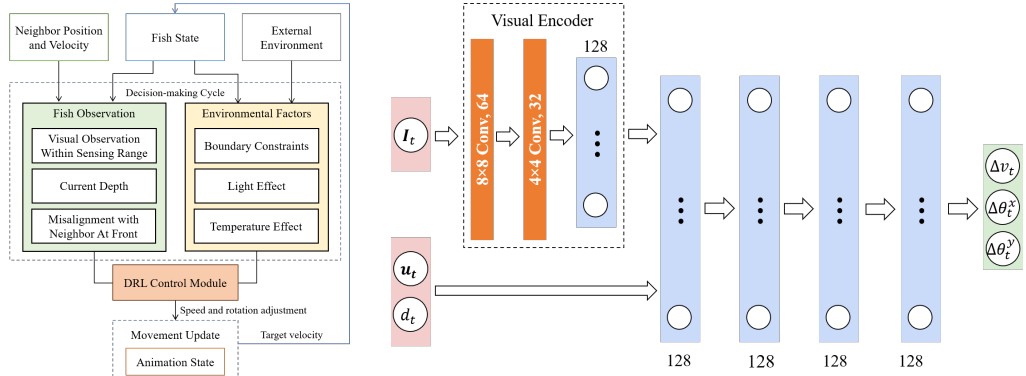

Figure 1: Simulation framework.    Figure 2: Architecture of policy network.

animation is generated by the finite state machine based on the velocity of the locomotion and the fish's behavior state, which is explained in more details in the Appendix.

**Biological and Environmental Factors:**    We incorporate important biological and environmental drivers of fish swimming patterns into the simulation framework. A fish stays within a comfortable distance of its neighboring fish and aligns its direction with that of its nearest neighbor in the forward direction [2, 41]. Its activity space is constrained by the cage boundaries and water surface, which reproduce the commercial cage environment and facilitate the formation of schooling. Additionally, caged fish are divided into dominant and subordinate groups, where the dominant individuals may initiate antagonistic acts by aggressively approaching the subordinate members. These factors are carefully integrated into the process of learning control policies that generate the delta velocity of fish $\Delta \mathbf{v}_t^f$ at each time step $t$ through the use of the PPO algorithm.

There are effects of underwater light intensity and temperature on the fish's vertical distribution, which we represent as $\Delta \mathbf{v}_{\text{light}}$ and $\Delta \mathbf{v}_{\text{temp}}$. A fish has a preferred range of light intensity and water temperature, and it changes its vertical position to stay in the comfortable zone. The details of the computation are explained in the Appendix.

In addition to the above-mentioned components, we integrate the fish decision making interval into the framework to emulate the latency of fish's responses to environmental changes. The duration (in units of simulation steps) of the decision making interval, $\Delta t_{\text{res}}$, for simulated species is predefined in accordance with the literature studies described in Sec. 2. At the time of simulation, given the time interval of simulation steps ($\Delta t_{\text{sim}}$), a fish updates its observation of the environment every $\lfloor \Delta t_{\text{res}}/\Delta t_{\text{sim}} \rfloor$ steps and takes actions between updates. The delta accumulated velocity to apply at each simulation step ($\Delta \mathbf{v}_t^a$) can then be derived as follows:

$$\Delta \mathbf{v}_t^a = \Delta \mathbf{v}_t^f + \frac{\Delta \mathbf{v}_{\text{light}} + \Delta \mathbf{v}_{\text{temp}}}{\lfloor \Delta t_{\text{res}}/\Delta t_{\text{sim}} \rfloor}. \tag{1}$$

**States and Actions:**    Each state $s_t \in$ state space $\mathcal{S}$ encodes the information a fish agent observes in the environment at time step $t$. It can be represented by a tuple $(\mathbf{u}_t, d_t, \mathcal{I}_t)$, where $\mathbf{u}_t$ is the difference between forward directions of the agent and its nearest neighboring fish in front of it at current step $t$, $d_t$ is the depth of the agent with respect to the water surface, and $\mathcal{I}_t$ is a visual observation tensor. During the time of exploration, a fish agent collects visual observations with a spatial grid sensor that imitates the sensing area of real fish [42]. The visual observation is stored as a third-order tensor, whose dimensions are grid width, grid height, and number of channels. The width and height are defined by the grid resolution, which is set to be 34×20. There are 6 channels encoding a scalar value of the normalized distance from the closest detected object within the fish's sensing range $d_{\text{sense}}$ to the agent and a one-hot encoding of the object type (i.e. fish, boundary or obstacle). $d_{\text{sense}}$ is valued at 2 body lengths (BL) for yellowtail amberjack (yellowtail) [43] and 3 BL for coho salmon and red seabream [42]. We stack three visual observations together to infer movement before passing them to the networks. All the state components are computed in the local coordinate system of the agent, with the origin located at the body center and the z-axis parallel to the fish's facing direction. Note

that a simulated fish is not capable of precisely observing its speed and rotation since a real fish can only sense its relative rotation through the use of the lateral line system [44, 45].

The action $a \in$ action space $\mathcal{A}$ determined by the policy specifies the delta speed ($\Delta v_t$), as well as delta rotation about the x-axis ($\Delta \theta_t^x$) and the y-axis ($\Delta \theta_t^y$) in degrees. The rotation angle about the z-axis is clamped blueto a small angle $\theta^{zt}$ to avoid unnatural rolling behavior. $\Delta v_t$ is also clamped by the maximum delta speed $\Delta v_{\max}$ allowed in the cage environment. The three action components are then used to compute the delta velocity of fish $\Delta \mathbf{v}_t^f$ at the current time step and thus drive the motion of the fish agent as depicted in Eq. (1).

**Reward:** The reward $r_t$ at each time step is defined to encourage schooling behavior while avoiding boundary collisions and to be consistent with the biological studies described in Sec. 2:

$$r_t = r_t^{BC} + r_t^{NC} + r_t^{BD} + r_t^{ND} + r_t^E + r_t^M + r_t^C. \tag{2}$$

The reward $r_t^{BC}$ represents the penalty of colliding with the spatial boundaries, which include the cage walls and water surface. It has a fixed value of $-300$ if a boundary collision occurs or 0 otherwise. $r_t^{NC}$ penalizes the collision with neighboring fish detected by box colliders using an associated weight $w^{NC}$ and accumulates with the number of colliding agents $N_{\text{hit}}$:

$$r_t^{NC} = -w^{NC} \, N_{\text{hit}}.$$

The boundary avoidance reward $r_t^{BD}$ encourages the fish to keep a distance from a detected spatial boundary. Its value depends on the agent's sensing range $d_{\text{sense}}$, the number of detected boundaries $N_{\text{bnd}}$, the distance $d_{\text{i}}$ to the boundary $i$ and the boundary avoidance weight $w^{BD}$:

$$r_t^{BD} = -w^{BD} \sum_{i=1}^{N_{\text{bnd}}} \left( \frac{d_{\text{sense}} - d_{\text{i}}}{d_{\text{sense}}} \right).$$

The neighbor interaction reward $r_t^{ND}$ encourages the fish to stay close to its neighbors within the sensing range and to align its direction with those of its neighbors. The angle $\Delta \theta_i^{\text{mov}}$ in degrees between the directions of the agent and each of its $N_{\text{nei}}$ neighbors is calculated and a neighbor interaction weight $w^{ND}$ is used for the computation:

$$r_t^{ND} = w^{ND} \sum_{i=1}^{N_{\text{nei}}} \left( \frac{90 - \Delta \theta_i^{\text{mov}}}{90} \right).$$

On the other hand, $r_t^E$ penalizes energy consumption of the fish while rotating its body or adjusting speed. It is computed from a rotation penalty weight $w^r$, a speed penalty weight $w^s$ together with the delta angle $\Delta \theta_t$ of the body rotation and the delta accumulated speed $\Delta v_t^a$ at the current time step:

$$r_t^E = -w^r \, \Delta \theta_t - w^s \, |\Delta v_t^a|.$$

The movement reward $r_t^M$ encourages the fish to swim faster than a minimum speed and penalizes sudden changes in depth caused by aggressive pitch motion (around the local x-axis). In the expression below, the variable $\theta^{rt}$ denotes the pitch angle threshold, $v^{st}$ is the speed threshold, $\theta_t^x$ is the current pitch angle, and $v_t^a$ represents the current accumulated speed:

$$r_t^M = \begin{cases} -10 & \text{if} \quad \theta^{rt} \leq \theta_t^x \leq (360 - \theta^{rt}). \\ 2 & \text{if} \quad -\theta^{rt} < \theta_t^x \leq \theta^{rt} \text{ and } v_t^a \geq v^{st}. \\ 0 & \text{otherwise.} \end{cases}$$

Lastly, $r_t^C$ denotes the chase reward which encourages aggression or escape behavior based on the social rank of the fish. Specifically, a dominant fish (aggressor) randomly starts a chase mode with a small probability $p_a$ and initiates an attack on its nearest subordinate neighbor (target). This triggers the subordinate being chased to start its escape mode and swim away from the aggressor. We reward the aggressor by a fixed large value if it collides with its target. This process can be expressed using the aggressor's accumulated velocity $\mathbf{v}_t^a$, a normalized vector $\mathbf{d}$ spanning from the aggressor to the target, a chase reward weight $w^{\text{agg}}$ for the aggressor, and an escape penalty weight $w^{\text{tar}}$ for the target as follows:

$$r_t^C = \begin{cases} w^{\text{agg}} (\mathbf{d} \cdot \mathbf{v}_t^a) & \text{if fish is an aggressor in chase mode and chases after its target.} \\ 100 & \text{if fish is an aggressor in chase mode and collides with its target.} \\ -w^{\text{tar}} (\mathbf{d} \cdot \mathbf{v}_t^a) & \text{if fish is a target in escape mode.} \\ 0 & \text{otherwise.} \end{cases}$$

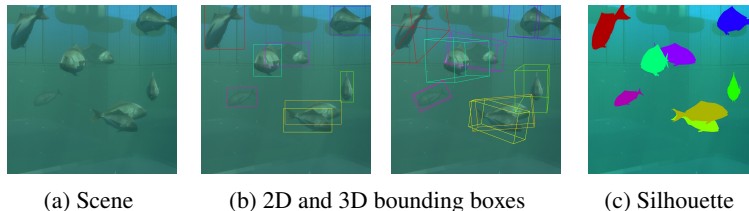

| (a) Scene | (b) 2D and 3D bounding boxes | (c) Silhouette |

Figure 3: Synthetic dataset example. (a) scene rendering. (b) 2D and 3D bounding box annotations. (c) silhouette annotation.

The chase and escape modes end for the two fish either when they collide, the target escapes to the back of the aggressor, or after a short period of time as the dominants prefer not wasting energy on a prolonged chase. A detailed explanation of parameters used in action definition and reward function is available in Appendix.

**Network:** The simulation of realistic fish schooling behaviors can be broken down into a series of tasks where fish agents need to choose optimal actions from continuous action space. In the Actor-Critic framework used by PPO, a policy (actor) $\pi$ is modeled as a neural network (Fig. 2) that maps a state $s_t$ to a Gaussian distribution over action $\pi(a_t|s_t)$. The agent's visual observation $\mathcal{I}_t$, which is formed as a 3D tensor, is processed by a visual encoder comprising two convolutional layers and a dense layer before being passed to a fully-connected network with other observation components. The first and the second convolutional layers contain 16 8×8 filters and 32 4×4 filters respectively and use Leaky ReLU activation. The fully-connected layer in the visual encoder contains 128 units and also uses Leaky ReLU as the activation function. Each of the other four fully-connected layers consists of 128 units and uses the Swish activation function. The output layer is just a linear layer with the output size equal to the size of action space. We train the policies using PPO with a clipped surrogate objective [39]. An in-depth review of this algorithm can be found in the Appendix. The advantage for the policy gradient is calculated by GAE [40]. The value function (critic) $V(s_t)$ is represented as another deep neural network with similar architecture except the output layer is a single unit. The networks are built on the open source library PyTorch.

## 4 Synthetic Dataset and Fish Counting

Our simulation is developed on Unity Engine [¶] with ML-Agents toolkit [46], and Crest Ocean System HDRP asset [‖] is utilized to create the effect of sun and ocean waves. The simulation is controlled by parameters including, but not limited to, fish biological details such as fish species, count, size, and speed; temperature and light preferences; latitude, longitude, date and time; wave size, sediment and chlorophyll concentrations; sunlight color and intensity; and light attenuation and scattering. A major drawback of prior fish simulation systems [31, 1] is the need for manual parameter tuning when applying the simulation to new species or new environments. However, in this work we leverage the power of DRL to train the fish to adapt to changing environmental conditions given their biological data, and apply the species-specific trained models to emulate the corresponding swimming patterns. We also use the physically-based environment simulation described in the Appendix to control the scene with a small number of hyperparameters instead of extensive adjustment with programmable shaders. These improvements significantly automate the simulation process and allow us to generate an arbitrarily large and varied dataset by randomly setting the above mentioned parameters within a physically valid range. Fig. 3 displays an example of a synthesized dataset.

The fish counting system comprises three modules: an image pre-processing module, which converts the input video to a sequence of images and applies denoising; a fish detection module, which is a trained network based on YOLOv4 [47] with the synthetic dataset; and a fish counting module. The system takes as input a fish cage video, and outputs an estimate of the total number of fish shown in the video. The details of each component can be found in the Appendix.

---

[¶]available at https://unity3d.com/

[‖]available at https://assetstore.unity.com/packages/tools/particles-effects/crest-ocean-system-hdrp-164158 under Standard Unity Asset Store EULA license

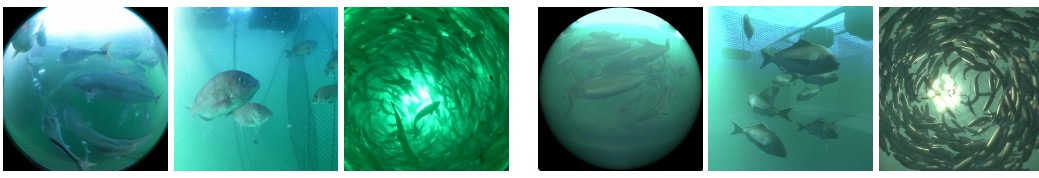

(a) Frames: real video            (b) Frames: simulation.

Figure 4: Comparison of the scenes captured from different views at a fish farm and in the simulation. Left-to-right: yellowtail, red seabream and coho salmon. Note that a fish-eye lens is used for yellowtail while a regular lens is used for coho salmon and red seabream.

## 5 Experiments and Results

We describe the training process, experimental results of the fish behavioral model with diverse biological and cage settings, underwater simulation, and fish counting.

**Training:** We trained the behavioral models of coho salmon, yellowtail and red seabream. A two-step training scheme was employed to facilitate the adaption for agents in eleven environments with the same action space, observation space and types of detectable objects, but different fish species, fish sizes, fish numbers, cage sizes and cage shapes. We pretrained the three species in their corresponding default scenes, which contain 1000 coho salmon, 45 yellowtail and 10 red seabream, respectively, with the cage configurations set to be the same as the fish farm environments we collected field data from. The pretraining results were usually sufficient to enable the agents to navigate and avoid cage boundaries to some extent, but collisions between fish and with boundaries still occur and unnatural movements persist. We then performed transfer learning based on the pretrained policies for each species in its corresponding environment-specific experiments, including the default scene that is used to reproduce the real scene. We found the use of such a two-step training scheme led to faster convergence and overall performance improvements compared to training from scratch for each environment. We enabled early termination of an episode if the agent collides with either a cage wall or the water surface. The agent would subsequently restart a new episode from a random position within the cage with a random valid rotation and an initial speed $v_0$. This strategy discourages boundary collision and prevents the agents from collecting data that is irrelevant to boundary avoidance. The values of hyperparameters in the pretraining and transfer learning phases are similar if not identical. For yellowtail and red seabream, we ran a total number of two million time steps and used a linearly decaying learning rate of 0.001 in both phases. For coho salmon, we adopted the same settings, but increased the pretraining and transfer learning steps to four million and one million time steps respectively, because the 1000 coho salmon in the default scene is much more than the other two species. The total wall-clock training times for four million and five million simulation steps were around 5500 seconds and 6800 seconds, respectively. A detailed explanation of the eleven simulation environments, hyperparameter settings, and the learning curves of two training stages are available in the Appendix. We also demonstrate the training progress of yellowtail in a recorded demo that is part of our supplementary video. The training took place on an NVIDIA GeForce RTX 3080 Laptop GPU installed on a Windows 10 machine with a 3.3 GHz AMD Ryzen 9 5900HX CPU.

**Fish Behavior:** We simulated swimming scenes of the above-mentioned three species in order to provide a variety of synthetic data to train the computer vision system that detects and tracks fish. Fish agents were initialized at random positions in the cage with random valid rotations, and would start to form circling patterns by following the trained policies. A qualitative comparison between the image sequences from the simulation and the videos captured underwater is displayed in Fig. 4. Note that the swimming patterns of the three fish species are distinct in the captured video. Our simulation successfully reproduced such behavioral differences as well as the environmental variation such as lighting, water color and turbidity.

We also showcase the simulation results of five example environments with different species, fish size, fish number, cage size and shape settings in Fig. 5. A body scale of 1.0 refers to the default body lengths (BL) of coho salmon, yellowtail and red seabream, which were set to be 0.49 meters, 0.52 meters, and 0.34 meters respectively according to the collected field data. We additionally applied a

mild repulsive force acting as an object-to-object colliding force to push any penetrating fish away from each other to further facilitate collision avoidance among agents. The fish swimming patterns vary with the scene configuration since they result from a combination of factors discussed in Sec 3. Notably, the 10 red seabream in Fig. 5(a: left 3) slowly circle around the cage center given the small cage size, whereas the same 10 fish in Fig. 5(a: right 3) learn to swim rapidly and away from the center since the cage size is much larger. On the other hand, the red seabream in Fig. 5(b) form a more compact school and slowly circle around the entire small cage because of their larger quantity. In addition, we simulated two fictional scenes where fish with large body size disparity (Fig. 5(c)) and multiple species (Fig. 5(d)) are mixed together. With associated biological measurements and trained policies, fish in each of the size and species groups exhibit distinct swimming patterns while influencing those of other groups in the same fish cage.

**Simulation vs Reality:**    We focus on the relationship between fish density in a cage and the resulting fish school states, and compare simulation results with data obtained in the field. We use two kinds of schools in our simulations, sparse ($0.59$ fish/m$^3$) and dense ($14.7$ fish/m$^3$), where the densities are set to be consistent with the field work. In the real fish farms, the sparse cage is $6.5$m $\times$ $6.5$m square with a depth of $6$m and contains $272$ fish. The dense cage is an octagon with $6.5$m edges and a depth of $10$m and contains about $30000$ fish. We quantify states of schools, and focus on milling, characterized by regular circulation around an axis, and swarming, in which fish aggregate with a weak polarization [48, 49]. Fig.6 shows sampled trajectories, a polar order parameter $P$ given by the norm of the average of direction vectors for each fish in a school, and a normalized angular momentum $M$ representing how regular the circulation is [49] for two schools with different densities. Details can be found in the Appendix. The simulation result demonstrates that the collective behavior of the school transitions from swarming to milling as density increases, and this tendency is in good agreement with the real data.  Several trajectories of individual fish in real video in both sparse and dense schools are exhibited in Fig. 7, and they resemble the ones in simulation results drawn by the bold trajectories in the right panels.

Moreover, we show the influence of the dominance hierarchy on fish behavior in Fig. 8. We initialized the scene with 20% of fish being dominant and the rest being subordinate based on the reported distribution of social hierarchies among fish in [38]. The dominant individuals would occasionally chase their subordinate neighbor until colliding with or losing sight of the latter, which matches our observation of the fish behavior in underwater videos captured in aquaculture sites. A demonstration of the simulation scenes is also included in our supplementary video.

**PPO vs SAC:**    We compare the results of simulations trained by PPO and SAC [50], in the default environment of coho salmon, in which 1000 coho salmon swim in a large octagonal cage with edges of 3 meters and a height of 4.6 meters. Both PPO and SAC eventually produced agents forming swimming patterns that resembled those of real coho salmon in reference videos, while the former yielded more stable outcomes. The same reward parameter setting used for PPO was initially applied to SAC in the pretraining stage. Although the SAC agents collected a higher average reward over episodes from an early stage of training than their PPO counterparts, they became over-trained in the later phase of training and exhibited a circling pattern where all salmon tended to cruise in the same direction and align their body rotations, which would be unnatural in a real-world farming cage setting. This result was jointly caused by a higher sample efficiency and entropy maximization strategy of the SAC algorithm that encourages exploration. After adjusting the reward parameters accordingly, SAC agents were able to swim in equally natural patterns as PPO agents after the pretraining and fine tuning stages. However, no significant improvement in the cumulative average reward was observed in the fine tuning stage of SAC and its resulting value was lower than that of PPO as shown in Fig. 9. A possible explanation is that the aggregated penalties from constraining factors such as boundary and neighbor collisions overwhelmed SAC agents as their off-policy nature required them to collect experience from previous episodes into the replay buffer for future updates.

**Underwater Simulation:**    The results of the experiments with each component of the environment simulation are illustrated in Fig. 10. Fig. 10(a) shows how the underwater scene becomes more reddish brown with higher concentrations of chlorophyll resulting in an increasing attenuation of blue. A higher value of sediment concentration makes the water cloudier as shown in Fig. 10(b). Fig. 10(c) depicts a timelapse of the scene. Fig. 10(d) demonstrates the change in brightness with increasing depth, since more light is absorbed at greater depths. Fig. 10(e) shows the influence of changing

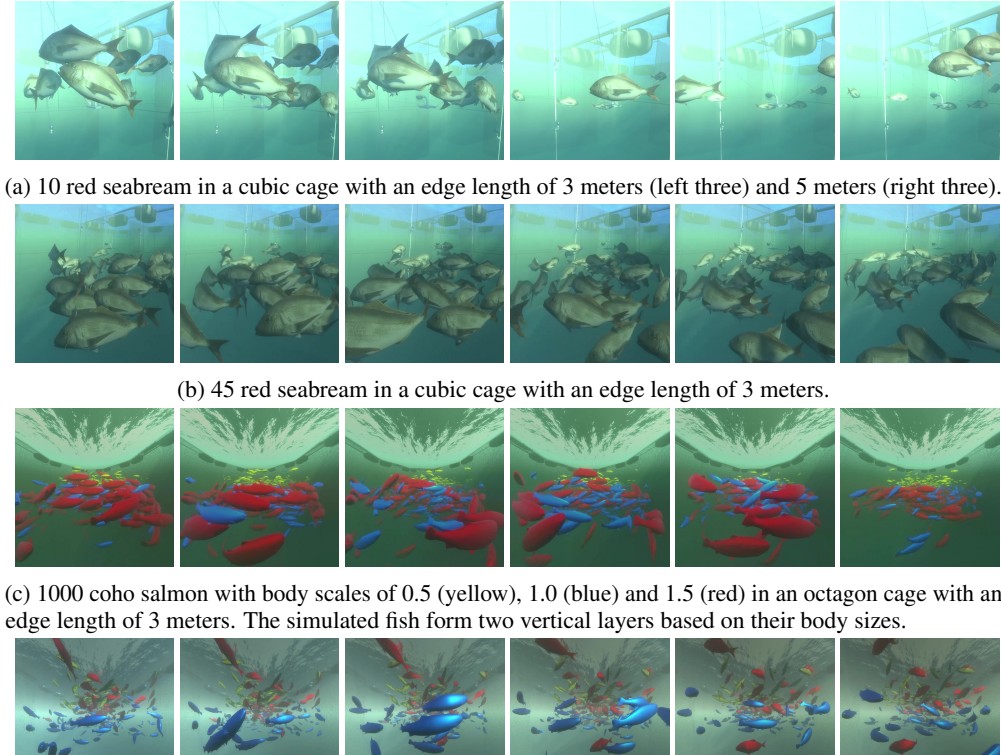

(a) 10 red seabream in a cubic cage with an edge length of 3 meters (left three) and 5 meters (right three).

(b) 45 red seabream in a cubic cage with an edge length of 3 meters.

(c) 1000 coho salmon with body scales of 0.5 (yellow), 1.0 (blue) and 1.5 (red) in an octagon cage with an edge length of 3 meters. The simulated fish form two vertical layers based on their body sizes.

(d) 100 coho salmon (blue), 100 yellowtail (yellow) and 100 red seabream (red) in an octagon cage with an edge length of 3 meters. The swimming pattern vary with their distinct biological factors.

Figure 5: Sequence of frames from example environments with diverse fish simulation configurations. All fish have random body scales within the interval [0.9, 1.1] by default unless specified.

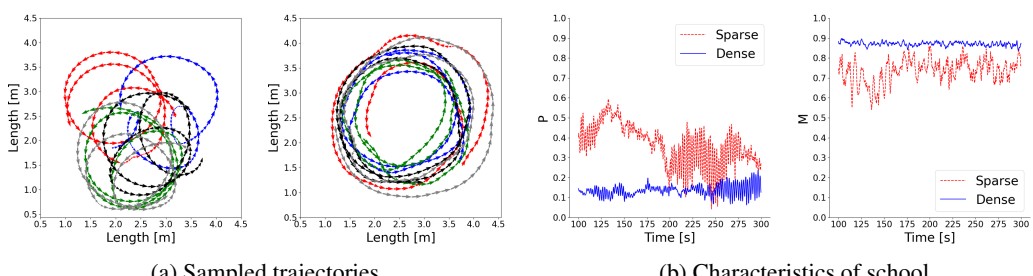

(a) Sampled trajectories                                    (b) Characteristics of school

Figure 6: Relation between density and states (swarming or milling) of fish schools and their characteristics: (a) 5 trajectories in sparse (0.59 fish/m$^3$, left) and dense (14.7 fish/m$^3$, right) schools. The left and right panels show swarming and milling state respectively. The fish cages are square with 4.0m edges in the left panel and octagon with 1.66m edges in the right panel. (b) Time series of $P$ (left) and $M$ (right) for dense (blue line) and sparse (red dashed line) schools.

small particle concentration on the brightness of the scene. A higher concentration of small particles increases the amount of light scattered, both towards and away from the camera, causing the light around the sun to brighten and everything away from the light source to darken. Fig. 10(f) shows the influence of large particle concentration, which works the same way as small particle concentration but has a smaller effect on light scattering.

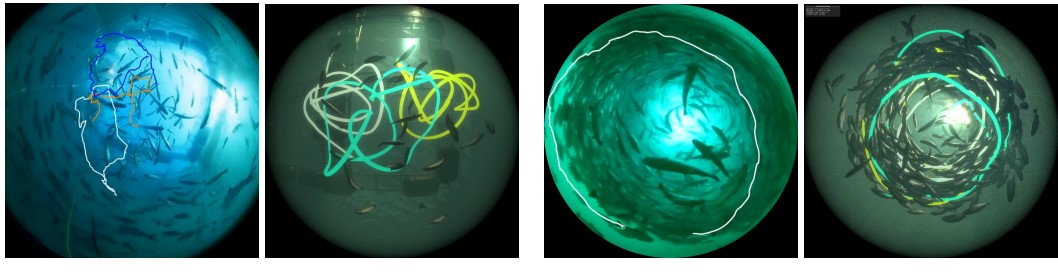

(a) Sparse school             (b) Dense school

Figure 7: Fish trajectories in schools in real video (left panels) of coho salmons and simulated ones (right panels). The number of fish and the size and shape of the fish cage in each simulation are described in the caption in Fig. 6. The parameters of real ones are as follows : (a) a sparse school (the fish count is 615) in a rectangular cage with 6.5m edge and 6m depth. Some fluctuations are observed on the orbits in real videos because the camera cannot be fixated completely in the ocean. (b) a dense school (the fish count is approximately 30000) in a octagon cage with 6.5m edge and 10m depth.

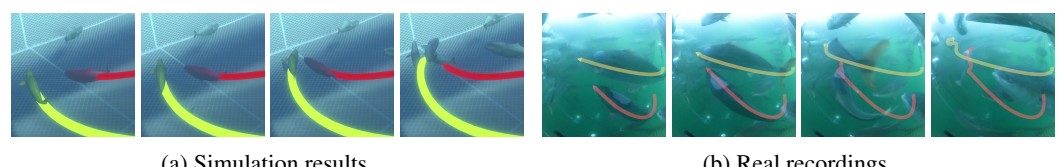

(a) Simulation results.             (b) Real recordings.

Figure 8: Sequence of frames from an aggressive behavior instance where a dominant fish (with red trajectory) chases after a subordinate neighbor (with yellow trajectory) until hitting the latter.

**Fish Counting:** We applied our modified YOLOv4 model [47]** trained with the synthetic dataset to real footage. Fig. 11 shows some results with the videos of coho salmon, yellowtail and red seabream. The trained fish counting algorithm estimated the number of fish in each frame, and the total number of fish was computed by taking the max of the estimated values while the network processes all the video frames. The counting results for coho salmons, yellowtails and red seabreams are 69:61, 22:32, and 7:7 (left:our system, right: manually counting) respectively. When the density of the fish is high and many occlusions are observed, the accuracy of the counting system drops.

## 6 Conclusions and Future Work

We presented a bio-inspired fish simulation system with DRL which adapts to various conditions autonomously by applying transfer learning, and which enabled us to simulate a broad range of fish behaviors in many scenarios. In addition, we introduced a physically-based underwater simulation which enabled us to simulate various undersea conditions. Using these, we were able to synthesize a large dataset with accurate annotations which we could use for computer vision tasks. We presented one application, fish counting, in this paper, but the potential of this synthetic dataset created with our novel simulations goes beyond fish counting. We will extend the application to 3D detection, pixel-wise segmentation, size estimation and tracking tasks in order to make more useful tools for fish farms.

A limitation of the current pipeline is the manual setting of weights for reward functions. We will explore techniques that can dynamically adjust weights based on the performance of the DRL controller in the training process. Moreover, we chose not to apply any behavior cloning method because it was difficult to collect and annotate enough samples of fish trajectory data from real video, which was the main challenge we aimed to solve with the proposed method for the preparation of the dataset for computer vision models. However, the fine-tuning of the trained DRL model using behavior cloning with a small sample of the real fish behavioral data is an interesting method to enhance the reality of the simulation that can be worth working on.

---

**under Apache License 2.0

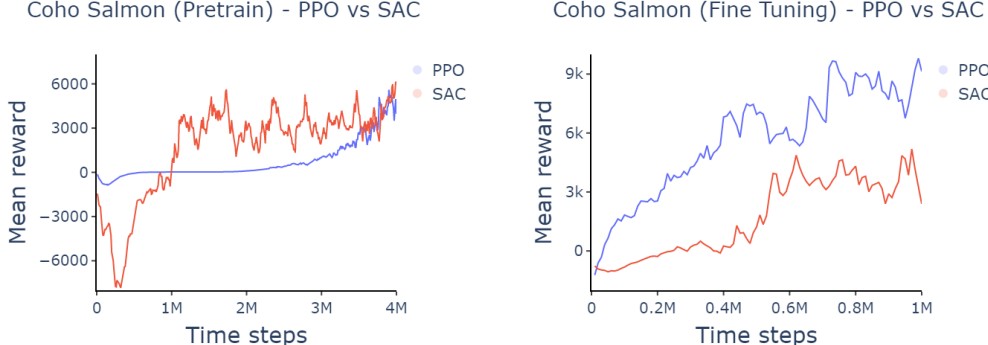

Figure 9: Comparison of per-episode mean cumulative rewards over all coho salmon agents trained using PPO (blue) and SAC (red) algorithms in pretraining (left) and fine tuning (right) phases.

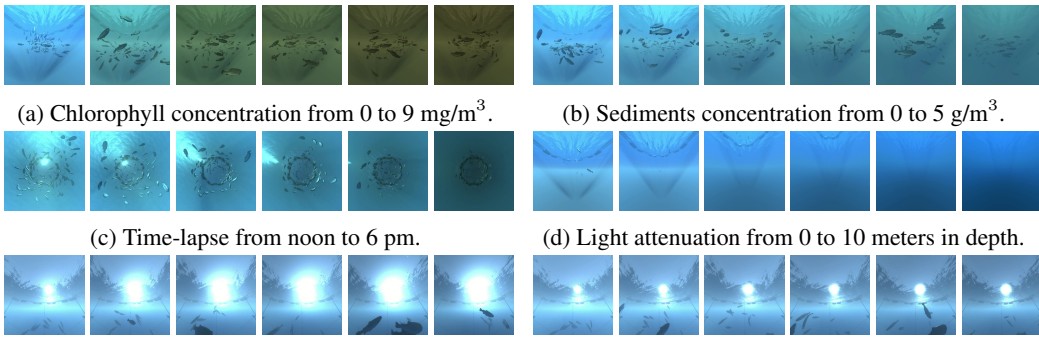

(a) Chlorophyll concentration from 0 to 9 mg/m$^3$. (b) Sediments concentration from 0 to 5 g/m$^3$.

(c) Time-lapse from noon to 6 pm. (d) Light attenuation from 0 to 10 meters in depth.

(e) Light scattering with small particles ($<$1 micron in diameter) concentrations from 0 ppm to 4 ppm. (f) Light scattering with large particles ($>$1 micron in diameter) concentrations from 0 ppm to 4 ppm.

Figure 10: Experiments with each component of the environment simulation.

In this paper, we excluded the effect of tidal currents and the influence of fish body temperature on movement dynamics from our simulations, and the proposed simulation method is not applicable to special occasions such as fish behavior in feeding and disease conditions yet. We plan to incorporate them to make our simulation more biologically accurate.

## Acknowledgment

We are grateful to K. Suda, T. Yoshida, G. Yasui and others from SoftBank Corp., R. Mizutani, T. Tsurumi, S. Sone, and others from Nosan Corporation for field works. We also thank M. L. Eastman and S. Kakazu from SoftBank Corp. for their supports on computer graphics pipeline and M. A. N. Kemas from NeuralX Inc. for computer vision models.

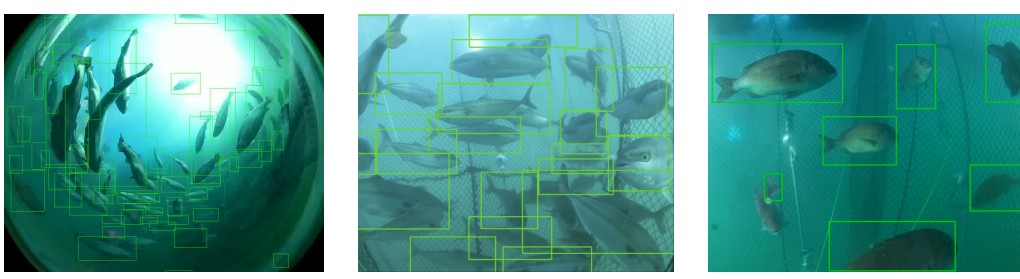

Figure 11: Example images of a fish counting system using YOLOv4 trained model. The captured video data with (left) coho salmon (middle) yellowtail (right) red seabream were fed to our fish counting pipeline, and the total number of fish in each cage was estimated automatically.

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
