# OpenReview forum: "DeepFoids: Adaptive Bio-Inspired Fish Simulation with Deep Reinforcement Learning"
_NeurIPS.cc/2022/Conference — NeurIPS 2022 Accept_

### Official Review · Reviewer_e9JA · 2022-07-06

**Rating:** 7
**Confidence:** 4
**Soundness:** 3 good
**Presentation:** 3 good
**Contribution:** 3 good

**Summary:**

The proposed paper aims at defining a fish simulation with deep reinforcement learning to mimic the behavior of swarms of fish. Fish behavior is simulated based on a multi-agent deep reinforcement learning approach and a combination of different rewards. Additionally, an underwater simulation is proposed to define plausible scenes. To validate the method, the simulation is used to generate training data for common computer vision tasks (e.g. counting of fish).

**Questions:**

- L113: What is meant by multi-hot values? I assume this refers to a multi-one hot vector? Please clarify.
- L116: What exactly are the 'visual observations'? The rendered images of three frames?
- L123: 'Clamped by a small angle'? When I am not mistaken this should be 'clamped to'.
- L152-154: While it is clear how the aggressors velocity is changed, I am missing a description of how the subordinates behavior is modeled. Put differently, what exactly is the 'chase mode' of the subordinate fish?

**Limitations:**

While Sections 5 and 6 briefly touch on the limitations of some aspects of the method (the accuracy of Yolo is actually not a limitation of the proposed method), I would encourage the authors to add a more in-depth discussion on the limitations of the proposed simulation. What are the failure cases of the simulation? What kind of fish behavior cannot be captured? etc. A discussion on the limitations could initiate future work in this direction. L314-316 also touch on a few topics but in a rather shallow manner. Here more detail would be appreciated.

**Strengths And Weaknesses:**

Strengths

- Simulating the behavior of fish with deep multi-agent reinforcement learning is novel and interesting.
- The paper is very-well written and easy to follow.
- The approach is technically sound and the results are impressive.
- The use of synthetic data for training neural networks for downstream tasks (counting fish) is convincing.
- The simulation is calibrated with field data.

Weaknesses

- The only negative point I have regarding this paper is that it does not quite seem a fit for a conference paper in terms of the page limit. A lot of interesting and important content is presented in the appendix (it is 14 pages long), so it seems the paper would be a better fit for a journal. However, given the high quality of the paper I think this should not be a reason for rejecting this work.

---

> ### Author Response · Authors · 2022-08-02
> **Thank you very much for the review. Here are our comments/ answers to the questions**
>
> We really appreciate all the positive feedback and constructive suggestions.
>
> - A lot of interesting and important content is presented in the appendix, so it seems the paper would be a better fit for a journal. However, given the high quality of the paper I think this should not be a reason for rejecting this work.
>
> Thank you very much for your kind understanding.
>
> - L113: What is meant by multi-hot values? I assume this refers to a multi-one hot vector? Please clarify.
> - L116: What exactly are the 'visual observations'? The rendered images of three frames?
>
> Yes, it should be phrased as “multi-one-hot vector”. Each visual observation is the detection result of the spatial grid sensor in a fish agent stored as a 3D tensor. The width and height of this tensor are defined by the grid resolution, which is set to be 34x20. The depth contains 6 channels encoding the scalar value of the normalized distance from the closest detected object to the agent and a one-hot encoding of the object type (fish, boundary, obstacle).Three observation tensors of three subsequent decision making steps (controlled by decision interval) are then stacked together along their depth, thus the concatenated tensor to feed to the network contains multi-one-hot encoding of detected object types. We will make sure to clarify this in our revision.
>
> - L123: 'Clamped by a small angle'? When I am not mistaken this should be 'clamped to'.
>
> Yes it should be “clamped to”. Thank you for the correction.
>
> - L152-154: While it is clear how the aggressor's velocity is changed, I am missing a description of how the subordinate's behavior is modeled. Put differently, what exactly is the 'chase mode' of the subordinate fish?
>
> We agree that the name “chase mode” is confusing. We will change it to “escape mode” for subordinate fish in our revision. A subordinate fish in an “escape mode” will just try to escape from the aggressor until the aggressor stops chasing, which will happen either after a short period of time or when the subordinate escapes from the sensing area of the aggressor.
>
> -  I would encourage the authors to add a more in-depth discussion on the limitations of the proposed simulation. What are the failure cases of the simulation? What kind of fish behavior cannot be captured? etc. A discussion on the limitations could initiate future work in this direction. L314-316 also touches on a few topics but in a rather shallow manner. Here more detail would be appreciated.
>
> Thank you for the feedback. We will add more in-depth discussion in the final version. We implemented a good amount of biology; however, behavioral models in feeding and in disease conditions are not implemented yet. Moreover, although we implemented the preference of the temperature in the fish simulation, the effect of the body temperature in the movement dynamics, which is the major characteristic in the thermoregulatory animals, has not been implemented. Because our fish was simply animated kinematically and not being controlled by a neuromuscular controller with the musculoskeletal system, the variations of movements caused by the factors such as muscle strength and the force propagation are still missing. Those are some of our future works.

---

### Official Review · Reviewer_RkxH · 2022-07-11

**Rating:** 5
**Confidence:** 3
**Soundness:** 3 good
**Presentation:** 3 good
**Contribution:** 2 fair

**Summary:**

The authors present a method for augmenting computer vision task with simulation, with specific application to fish simulation. They create a simulation environment using deep reinforcement learning to mimic the behavioural properties of several fish species. The simulated behaviour is then rendered in a physically realistic computer-generated environment. Finally, computer vision modules are used downstream to, for example, identify and count fish.

**Questions:**

Is it possible to perform some clustering on the video data and compare clustered behaviours/features to those generated using the simulation? This would give a much easier way for assessing how successful this approach has been.

The use of simulated synthetic data to improve downstream video tasks could be a useful tool for assessing problem cases where data is sparse/collection if expensive. For example, the synthetic data could be used to identify adversarial examples where, in this example, the fish counting algorithm performs poorly. Have the authors considered this?

Is it possible to closely couple the simulation with the available data, for example, through in-filling trajectories of fish videos given initial trajectories? However, I appreciate these agent-based simulators are highly stochastic and this may not be meaningful in practice.

**Limitations:**

This work is very closely centred on simulating fish behaviour, aided by deep learning. The manuscript is currently structured such that it is more relevant to modelling fish behaviour than advances in machine learning. While I believe that this work represents a significant advance in reproducing the collective dynamics of fish species, the authors should consider how to structure and emphasise this work such that it has more relevance to advances in deep reinforcement learning.




**Strengths And Weaknesses:**

The authors provide a comprehensive and detailed simulation of fish behaviour. This is then optimized using deep reinforcement learning. They apply this approach to problems in fish farming, which they sufficiently justify in their introduction. While this work is original, with respect to the combination of many different approaches to their task problem, it is not entirely clear what is novel in the learning methods.

In particular, the authors deal with three main problems which they address with machine/deep learning (simulation of fish behaviour, rendering fish behaviour in photo-realistic videos, identifying fish in videos). However, there is no attempt to unify learning across these problems in any significant way, so they remain relatively disjoint. Moreover, it is very difficult to judge the performance of the deep learning approach as the quantified assessment is done at the scale of fish identification, as discussed below.

The main result of this work is the accurate simulation of fish behaviour, using deep learning. However, the presented summarising features separating out different simulation environments, swarming behaviour and change in polarity and momentum over time, are compared only against some subsets as example. These are not sufficient to judge whether this work is successful at capturing fish behaviour. I suggest the authors consider quantified metrics for the ground truth data they have on fish behaviour and systematically compare this to the simulated behaviours.

---

> ### Author Response · Authors · 2022-08-02
> **Thank you very much for the review. Here are our comments/ answers to the questions.**
>
> Thank you very much for the review.
>
> - While this work is original, with respect to the combination of many different approaches to their task problem, it is not entirely clear what is novel in the learning methods.
> - Emphasis on more relevance to advances in deep reinforcement learning.
>
> The motivation of this paper is to leverage the power of DRL to achieve realistic fish simulation that can generate high-quality synthetic dataset with various environment settings without high cost of underwater data collection nor the manual scripting for the animation. The novelty lies in our design of the whole system such that it has the backing of fish biological studies and yields simulation results that closely resemble real-world observations. This concept of the biologically inspired reward/ penalty design can be further generalized to facilitate the learning effectiveness of other RL models where domain knowledge is helpful.
>
> - There is no attempt to unify learning across these problems (simulation of fish behavior, rendering fish behavior in photo-realistic videos, identifying fish in videos)
>
> We did not need to unify the three components to achieve our goal, which was to synthesize enough variation of the underwater scenes to train computer vision models. It is going to be interesting to unify the 3 components and keep improving the synthetic dataset framework and the trained computer vision model together as the recurrent loop to fine-tune the quality of the simulation as well as the accuracy of the computer vision models in the unified pipeline.
>
> -  It is very difficult to judge the performance of the deep learning approach as the quantified assessment is done at the scale of fish identification. The presented summarizing features separating out different simulation environments, swarming behavior and change in polarity and momentum over time, are compared only against some subsets as example. These are not sufficient to judge whether this work is successful at capturing fish behavior.
> - I suggest the authors consider quantified metrics for the ground truth data they have on fish behavior and systematically compare this to the simulated behaviors.
> - Is it possible to perform some clustering on the video data and compare clustered behaviors/features to those generated using the simulation?
>
> It is difficult to obtain the fish pose and trajectories in the real-world to conduct more quantitative analysis between the simulation result and the real-world data. LiDAR sensors and IR sensors cannot be used due to the rapid damping of signals in the water. Furthermore, a sonar used in a fish finder cannot be used in the fish cage because the fish is too close to the sensor. Because of the lack of the capturing methods in the real-world scenes, it is not easy to execute the systematic comparison with larger quantities of the dataset to evaluate whether the quality of the resulting simulation is good or not. The difficulty of the data capture is exactly the reason why we needed to implement the proposed data synthesis approach which can generate enough dataset to train computer vision models. Another way to obtain the trajectory/ pose data for the quantitative analysis or the clustering analysis is to apply a computer vision model onto captured real video frames. In fact, that is what we are trying to develop by training neural networks with the dataset generated by the proposed synthetic data platform.
>
> - The use of simulated synthetic data to improve downstream video tasks could be a useful tool for assessing problem cases where data is sparse/collection if expensive. For example, the synthetic data could be used to identify adversarial examples where, in this example, the fish counting algorithm performs poorly. Have the authors considered this?
>
> Yes, those are great points. We believe that we can improve the quality of the dataset by taking the suggested approach and enhance the accuracy and robustness of the trained computer vision models. It is one of the advantages that simulation can offer, where the trained computer vision model can be easily tested to find the failure cases whereas it is quite challenging to conduct such testing in the real world.
>
> - Is it possible to closely couple the simulation with the available data, for example, through in-filling trajectories of fish videos given initial trajectories? However, I appreciate these agent-based simulators are highly stochastic and this may not be meaningful in practice.
>
> The goal of the current proposed method is to synthesize various enough scenes so that a comprehensive dataset can be fed to computer vision training. When the number of the fish and the size / shape of the fish cage as well as other environmental factors change, the dynamics of the system change quite a lot, and the resulting trajectory for each fish changes. It is almost impossible to collect all the trajectory data in every scenario. That is why we took the highly stochastic approach.

---

> > ### Comment · Reviewer_RkxH · 2022-08-09
> > **Response to the Authors**
> >
> > Thank you for taking the time out to respond to my review. Having considered your response and manuscript, I have decided to update my rating.

---

> > > ### Author Response · Authors · 2022-08-09
> > > **Thank you very much for the update.**
> > >
> > > Thank you so much for taking the time to read our comments and being open-minded to hear them out. We really appreciate that you updated the final rating.

---

### Official Review · Reviewer_ZV3m · 2022-07-14

**Rating:** 6
**Confidence:** 4
**Soundness:** 2 fair
**Presentation:** 3 good
**Contribution:** 3 good

**Summary:**

This paper proposes a multi-agent reinforcement learning algorithm to simulate the shoaling/schooling behavior of fish. The main application of the method is to control position and movement of fish in a 3D simulation which enables synthetic data generation for downstream computer vision tasks such as counting and sizing.

**Questions:**

What RL implementation was used? I don't think this was mentioned in the paper.

Have the authors thought about domain randomization? I.e. randomizing textures, 3D models of fish and other parameters of data generation pipeline and how does it affect the final accuracy?

**Limitations:**

Authors briefly addressed limitations in the conclusion - from the data presented it is hard to judge the applicability of this method beyond this one task.


**Strengths And Weaknesses:**

This paper presents an interesting practical application of reinforcement learning - simulating animal behavior for synthetic data generation.
It is great to see Deep RL applied to a practical problem! The results in the paper also seem pretty convincing: some of the rendered images from the simulation seem to closely resemble real world images.

My first point of criticism is lack of support for the claims made in the paper. Section 1 suggests that using RL for synthetic data generation in this scenario improves the accuracy of downstream vision models. The accuracy of the system is evaluated only on three videos and is never compared to anything. Typically a machine learning paper that proposes a method should compare this method's results with a baseline of some sort to support the claim that it's a good method to use for such task. Here obviously RL algorithm would be expensive to run and tune and it's advantages are not apparent unless you compare the results with obvious baselines, such as for example running Foids/Boids algorithms and comparing performance of resulting models, or even placing fish in the simulated cage randomly.
While using RL for this task definitely looks interesting, I'm not convinced that it will significantly improve the results of the final model (I would be convinced if such measurements were present in the paper).

Authors also mention that they "attempt to automate fish counting and sizing" although the sizing task is never mentioned in the paper again. I think it would have been more correct to say that this method can also be used for other task, rather than suggesting that something has been attempted but never showing the actual outcome of the attempt.

To avoid being too negative in my review, it must be said that the final results are absolutely fantastic. The supplementary video is absolutely great too. Appendix is very comprehensive and contains a lot of valuable information, especially domain expertise.
I am conflicted in my evaluation, because this is clearly a high-profile and important work addressing an important ecological problem and this paper can have a great impact in its sub-area. At the same time it lacks some of the essential elements of a machine learning publication. I think the potential impact of this paper outweights its weaknesses therefore I'm leaning towards accepting this paper.

---

> ### Author Response · Authors · 2022-08-02
> **Thank you very much for the review. Here are our comments/ answers to the questions.**
>
> We really appreciate the positive feedback and constructive suggestions.
>
> - Lack of support for the claims made in the paper: RL’s advantage in generating synthetic data to improve vision model performance is not apparent unless the result of vision model is compared against baselines, such as the results produced by vision models trained using dataset generated by Foids/Boids algorithms or placing fish in the simulated cage randomly.
>
> Thank you very much for the suggestions. Yes, we conducted the experiments by placing the DRL trained fish in various fish cages randomly. We will include the ones with the Foids/ Boids model in the final version of the paper if the extra space is allowed. We agree that such baseline comparisons are better to be included.
>
> Even with the Foids/Boids algorithms, we can train the computer vision models to perform good enough computer vision tasks if we are willing to spend months of work to tune the parameters of the Foids/ Boids and generate a large enough dataset; however, the process is extremely tedious and time consuming. That is the reason why we proposed the current DRL based learning system to make the system to be adaptive to any scenarios to enhance the variability of the dataset without human manual tuning. We can show the difference of the trained computer vision model with the small and large dataset which are generated by Boids, Foids and the newly proposed DRL based method.
>
> - Authors also mention that they "attempt to automate fish counting and sizing" although the sizing task is never mentioned in the paper again.
>
> That is a good point. We will revise the paper accordingly.
>
> - What RL implementation was used?
>
> The PPO and SAC algorithms were implemented using the PyTorch framework. The details of the network can be found in Line 162-176 in the main paper. The training configuration of each method can be found in Appendix E.
>
> - Have the authors thought about domain randomization? I.e., randomizing textures, 3D models of fish and other parameters of data generation pipeline and how does it affect the final accuracy?
>
> We have already started working on introducing more variations of fish textures and models with manual tuning, and plan to further extend it by applying generative algorithms to synthesize the new textures and 3D models. Although the current pipeline already made the system work well enough, we think that the domain randomization will contribute to improve the accuracy of the fish detection and sizing estimation because the rendered scenes will be further diversified.

---

### Official Review · Reviewer_vrph · 2022-07-17

**Rating:** 6
**Confidence:** 3
**Soundness:** 3 good
**Presentation:** 3 good
**Contribution:** 2 fair

**Summary:**

The given work develops a fish simulation motivated by solving real world problems in the fish farming industry such as over and underfeeding. Through this work, they wish to replicate fish schooling behaviour with higher fidelity, by leveraging Deep Reinforcement Learning to model this. The authors argue that this resulting setup could then be used for training computer vision models to accurately count varied fish species in the cages.

**Questions:**

Section 4: Could details on the infrastructure to inject a Deep RL agent with the Unity environment be expanded? What is the latency of the system? Could researchers tune parameters of the simulation to allow for generalization across different scenarios?

Line 127-161: On a high level, could there be a description of the weighting of the scalar reward term? At the moment they all are considered to be equal.

   How is the upper bound of r_{t}{BC} computed in the scenario a collision takes place?

   How is wBD empirically computed? Is it a static parameter for a given experiment instance?

   It seems like a significant part of the intended behavior is being encoded through constraints.  Given the complexity of these   intricate behaviors, It would be interesting if this could be implicitly  learnt through forms of behaviour cloning, ranking relevant trajectories.






**Ethics Review Area:**

["I don’t know"]

**Limitations:**

Yes

**Strengths And Weaknesses:**

* Overall, the work is well written, well organized and clear to follow. It reasonably describes the past work in the field, stating how the contributions are distinct

* The authors motivate the problem statement well, on how this is an important societal problem to solve.

* However, the major contribution of the work focusses on designing this synthetic environment, which seems to be technically significant. What would add strength to this claim would the extension of the Simulation vs Real section, expanding on how this synthetic dataset translates to real world impact There is no major algorithmic novelty introduced. Details on the reward engineering could be expanded, since that seems an essential component of the work.

* Additionally, discussion on how this work could be more generalizable to the Reinforcement Learning/Computer Vision community would be useful. There are no details if the key contributions could be adapted in other domains.

---

> ### Author Response · Authors · 2022-08-02
> **Thank you very much for the review. Here are our comments/ answers to the questions.**
>
> Thank you very much for the review.
>
> - Extension of the Simulation vs Real section, expanding on how this synthetic dataset translates to real world impact.
>
> We demonstrated in the paper that the sparse school formed a swarming state while the dense school formed a milling state. In the appendix, we also showed the comparison of the real tracked trajectories and simulated ones in Fig. 2 and the distribution of the fish from the bottom view for the real and simulation one in Fig. 3 in the appendix. We will try to include more comparisons in the final version.
>
> - No major algorithmic novelty introduced.
>
> The novelty of our work is not in the DRL algorithm itself but lies in the design of the simulation system using DRL with the bio-inspired reward/ penalty setting and the physics-based environment simulation which enabled us to synthesize realistic scenes without manual scripting. We propose this approach to inspire researchers who work on machine learning problems where the data collection/ annotation is difficult to conduct. We proved that the resulting synthetic dataset became realistic in any given environment when we designed the system to be truthful to biology and physics and applied the DRL based learning system to enable the adaptation of the behavioral model to a given environment.
>
> - Details on the reward engineering.
>
> The reward function is designed based on fish biology studies, which are discussed in Appendix A. Appendix E provides more details regarding the values of reward function parameters that were set relative to the impact on the fish behavior as explained in Appendix A.
>
> - How could this work be more generalizable to the Reinforcement Learning/Computer Vision community?
>
> We demonstrated that the DRL framework with the biological constraints can yield simulation results that closely resemble real-world observations without the expensive manual trajectory planning under different scenarios. This concept can be generalized to synthesizing other animal behaviors in the wild. Although we have not tested the hypothesis in the other domains, we hope that our approach can inspire researchers who are in the area of computer vision and reinforcement learning to overcome their problem by taking a similar method, where scenes contain animals and available real behavior data is limited.
>
> - Could details on the infrastructure to inject a Deep RL agent with the Unity environment be expanded?
>
> DRL algorithms were integrated with Unity via Unity ML-Agents Toolkit. When there are less than 100 fish in the scene, running a trained DRL model has no latency, the frame rate can reach 40 FPS or higher. When there are 200 fish, the frame rate drops to 15-25 FPS. We used a Windows 10 machine with a NVIDIA GeForce RTX3080 Laptop GPU and a 3.3 GHz AMD Ryzen 9 5900HX CPU for the simulation. Because our fish agent controller is learning-based without manual scripting, the system is generalizable to any scenario by adjusting simulation settings and fine-tuning the trained model (see Appendix C for more details).
>
> - On a high level, could there be a description of the weighting of the scalar reward term?
>
> The weight of each reward term is adjustable, and it should be set according to the objective of each training. For example, we set the boundary avoidance weight to be the highest when we train the DRL controller for the very first time because the primary goal of the training for the fish was to avoid the collision to the fish cage. When it is learnt, we increased the weights for the neighbor collision penalty and energy consumption penalty so that fish learn the smooth schooling in an environment. The details are found in Table 3 of Appendix E.
>
> - How is the upper bound of r_{t}{BC} computed?
>
> r_{t}{BC} is empirically determined to be a large negative value at the time of collision to facilitate the learning of boundary avoidance.
>
> - How is wBD empirically computed? Is it a static parameter for a given experiment instance?
>
> w_{BD} is a static parameter (as shown in Table 3 of Appendix E) in our experiments. It is currently empirically determined by trial and error. It can be interesting to make it to be dynamic, where the weight value for each reward gets adjusted depending on the performance in the training process. We will think about good performance measurement of the trained DRL controller and the logic to update the weights dynamically.
>
> - It would be interesting if the intended behavior could be implicitly learnt through forms of behavior cloning, ranking relevant trajectories.
>
> We did not apply the behavior cloning nor imitation learning methods because it was difficult to collect and annotate enough samples of fish trajectory data, which was the main challenge we wanted to solve with the proposed method. However, fine-tuning the trained DRL using the behavior cloning with a small sample of the real fish behavioral data is an interesting method to further enhance the reality of the simulation.

---

> > ### Comment · Reviewer_vrph · 2022-08-06
> > **Thanks for addressing my comments**
> >
> > I've taken measure of the updated comments, reviewed my feedback and updated it.

---

> > > ### Author Response · Authors · 2022-08-07
> > > **Thank you very much for the update.**
> > >
> > > We really appreciate it that you spent the time on reviewing our comments and upgraded the rating. We are very grateful.

---

### Meta-Review · Area_Chair_8Cn3 · 2022-08-20

**Recommendation:** Accept
**Confidence:** Certain

**Metareview:**

This paper proposes to build photorealistic 3D models and simulation of fish schools in fish cages, with individual fish represented by swimming agents that are trained using multi-agent deep RL with bio-inspired rewards. The authors use these trained models, tuned to different fish species, to generate photorealistic images that will be fed to image YOLO-based detectors and counters, for applications to ecology and sustainable fishing. The trained fish detectors and counters is finally evaluated on real recordings.

Reviewers praised the clarity of the paper (vrph, e9JA), the motivation (vrph, ZV3m, RkxH, e9JA), the successful application of deep RL to a real problem (ZV3m, e9JA), the quality of the results (ZV3m, e9JA).

Reviewer vrph noted that the sim2real evaluation was insufficient, but the authors provided more details about comparisons of fish trajectories and distributions in the appendix. Reviewer ZV3m noted that no baseline (including Boids or random fish placement) had been used to train the fish detector and counter, and that the sizing task was mentioned but not done. Reviewer RkxH deplored the lack of metrics to qualify the learned fish behaviour. Reviewer e9JA noted that the paper was a bit long, with essential details in the appendix.

Reviewers agree on high scores (5, 6, 6, 7) and therefore I would recommend this paper for acceptance.

Thank you, Sincerely, Area Chair

**Award:**

No

---

### Decision · Program_Chairs · 2022-09-14

Accept